Effects of 12-week cadence retraining on impact peak, load rates and lower extremity biomechanics in running

Wang Junqing 1
Luo Zhen 1
http://orcid.org/0000-0002-1871-5886 Dai Boyi 2
Fu Weijie 1 3 fuweijie@sus.edu.cn
1 School of Kinesiology, Shanghai University of Sport , Shanghai , China
2 Division of Kinesiology and Health, University of Wyoming , Laramie, WY , USA
3 Key Laboratory of Exercise and Health Sciences of Ministry of Education, Shanghai University of Sport , Shanghai , China
Doyle Tim
Electronic publication date: 2020 Aug 24
Publication date: 2020
Volume: 8
Electronic Location ID: e9813
Received 2019 Jul 29; Accepted 2020 Aug 4
Copyright: © 2020 Wang et al.
Copyright year: 2020
Copyright holder: Wang et al.
License: This is an open access article distributed under the terms of the Creative Commons Attribution License, which permits unrestricted use, distribution, reproduction and adaptation in any medium and for any purpose provided that it is properly attributed. For attribution, the original author(s), title, publication source (PeerJ) and either DOI or URL of the article must be cited.
License URL: https://creativecommons.org/licenses/by/4.0/

Keywords: Running, Cadence, Impact peak, Lower extremity biomechanics

Funding: National Natural Science Foundation of China 11772201 and 11932013 National Key Research and Development Program of China 2019YFF0302100 “Dawn” Program of Shanghai Education Commission 19SG47 Talent Development Fund of Shanghai Municipal 2018107 Scientific Research Program of Shanghai Administration of Sports 20Q004 High-Level Internationalized Talents Program of Shanghai University of Sport This work was supported by the National Natural Science Foundation of China (11772201 and 11932013), National Key Research and Development Program of China (2019YFF0302100), the “Dawn” Program of Shanghai Education Commission (19SG47), the Talent Development Fund of Shanghai Municipal (2018107), Scientific Research Program of Shanghai Administration of Sports (20Q004), and the High-Level Internationalized Talents Program of Shanghai University of Sport. The funders had no role in study design, data collection and analysis, decision to publish, or preparation of the manuscript.

==============================
Background

Excessive impact peak forces and vertical load rates are associated with running injuries and have been targeted in gait retraining studies. This study aimed to determine the effects of 12-week cadence retraining on impact peak, vertical load rates and lower extremity biomechanics during running.

Methods

Twenty-four healthy male recreational runners were randomised into either a 12-week cadence retraining group (n = 12), which included those who ran with a 7.5% increase in preferred cadence, or a control group (n = 12), which included those who ran without any changes in cadence. Kinematics and ground reaction forces were recorded simultaneously to quantify impact force variables and lower extremity kinematics and kinetics.

Results

Significantly decreased impact peak (1.86 ± 0.30 BW vs. 1.67 ± 0.27 BW, P = 0.003), vertical average load rates (91.59 ± 18.91 BW/s vs. 77.31 ± 15.12 BW/s, P = 0.001) and vertical instantaneous load rates (108.8 ± 24.5 BW/s vs. 92.8 ± 18.5 BW/s, P = 0.001) were observed in the cadence retraining group, while no significant differences were observed in the control group. Foot angles (18.27° ± 5.59° vs. 13.74° ± 2.82°, P = 0.003) and vertical velocities of the centre of gravity (CoG) (0.706 ± 0.115 m/s vs. 0.652 ± 0.091 m/s, P = 0.002) significantly decreased in the cadence retraining group at initial contact, but not in the control group. In addition, vertical excursions of the CoG (0.077 ± 0.01 m vs. 0.069 ± 0.008 m, P = 0.002) and peak knee flexion angles (38.6° ± 5.0° vs. 36.5° ± 5.5°, P < 0.001) significantly decreased whilst lower extremity stiffness significantly increased (34.34 ± 7.08 kN/m vs. 38.61 ± 6.51 kN/m, P = 0.048) in the cadence retraining group. However, no significant differences were observed for those variables in the control group.

Conclusion

Twelve-week cadence retraining significantly increased the cadence of the cadence retraining group by 5.7%. This increased cadence effectively reduced impact peak and vertical average/instantaneous load rates. Given the close relationship between impact force variables and running injuries, increasing the cadence as a retraining method may potentially reduce the risk of impact-related running injuries.

Introduction

Long-distance running is a very popular form of physical activity in China and across the world (Clermont et al., 2019). According to the Chinese Athletics Association Marathon Annual Press Conference, 5.83 million participants ran in 1,581 marathon events (5k,10k, half and full marathon) in China in 2018 (Chinese Athletics Association, 2019). Similarly, 18.1 million runners registered for organised races in the US (Running USA, 2019). However, the incidence of running injuries is fairly high (Messier et al., 2018). A total of 19.4–79.3% of long-distance runners experienced lower extremity injuries (Van Gent et al., 2007). Amongst these injuries, knee injuries, such as patellofemoral pain, are the most common. Meanwhile, there is data to suggest that injuries to the lower leg have been reported to be just as common as injuries to the knee (Buist et al., 2010; Franke, Backx & Huisstede, 2019).

Training history, anatomical characteristics and running biomechanics are the main risk factors influencing overuse injuries due to running (Hreljac, 2004). Amongst various biomechanical factors, excessive impact peak forces and load rates are associated with injuries and have been targeted in gait retraining studies (Cheung & Davis, 2011). In a recent review, excessive accumulation of impact peak forces in knee joints was found to lead to overuse injuries (Gijon-Nogueron & Fernandez-Villarejo, 2015). Previous prospective studies found that injured runners had greater vertical load rates than non-injured runners (Davis, Bowser & Mullineaux, 2016; Fu et al., 2017). Another prospective study showed that runners with patellofemoral pain exhibited lower impact loading after the pain and disability typically associated with these injuries were reduced (Cheung & Davis, 2011). Impact peak can be influenced by several factors, such as speed (Hamill et al., 1983), shoe/surface/slope (Dixon, Collop & Batt, 2000; Gottschall & Kram, 2005), strike pattern (Davis, Rice & Wearing, 2017) and cadence/step length (Hobara et al., 2012). Increasing running cadence at 2.5 m/s or decreasing step length at 4.58 m/s could decrease impact peak and vertical load rates (Hobara et al., 2012; Stergiou, Bates & Kurz, 2003), and reductions in impact peak were related to decreases in vertical velocity of the centre of gravity (CoG) (Derrick, Hamill & Caldwell, 1998). Other lower extremity variables, such as foot angles at initial contact (Heiderscheit et al., 2011) and peak joint angles during the stance phase (Dos Santos et al., 2016), also showed decreases with increasing cadence. These results indicate that increasing cadence or decreasing step length has an effect on decreasing impact forces and other lower-extremity variables in running.

With regards to cadence retraining, Hafer et al. (2015) observed significant decreases in load rates after 6 weeks of cadence retraining with a 10% increase in cadence. However, cadence increased by only 2.4% after retraining, whilst a 10% increase in cadence was prescribed for retraining; no feedback was given as to how well the participants matched their prescribed cadence during retraining. Whether impact peak would decrease after supervised cadence retraining remains unknown. Therefore, a relatively long-term and supervised intervention is needed to evaluate the effects of cadence retaining on impact peak and load rates. Twelve weeks of gait retraining allows the initial adaptation of musculoskeletal structures to new running patterns (Latorre-Román et al., 2019) and may reduce injury risks of gait transition within a short period (Goss & Gross, 2012). Increasing one’s cadence by over 10% could be metabolically costly, which indicates that considerable increases in cadence are unsustainable and may not be adopted by runners (Cavanagh & Williams, 1982). Mobile monitoring was used by Willy et al. (2016b) to assess adherence to the prescribed cadence during cadence retraining with a 7.5% increase in cadence, and significant reductions in maximum and average load rates were observed after retraining. As such, we sought to determine whether a relatively small increase in cadence (7.5%) during long-term cadence retraining could significantly reduce impact peak and load rates.

The present study, therefore, aimed to quantify the effects of a 12-week cadence retraining protocol on impact peak, load rates and other lower-extremity biomechanical variables. We hypothesised that 12-week cadence retraining would result in remarkably decreased impact peak and load rates. Additionally, decreases in lower-extremity biomechanics at initial contact and during the stance phase after cadence retraining would be observed.

Methods

Participants

Sample size estimation indicated that a minimum sample size of 26 participants was required to achieve a minimum effect size of 0.6. Considering a drop-out rate of 15–20%, 30 male recreational runners were recruited through online social media, running clubs and flyers. Participants were randomly assigned to either a cadence retraining group (CAD) or a control group (CON) on the basis of the lottery method of sampling, and 15 participants were included in each group (Table 1). When the participants first reported to the laboratory, they were required to run on a treadmill for 2 min. A high-speed camera placed next to the treadmill recorded their foot strike patterns. The participants were determined to be rearfoot strikers by checking the foot angle (i.e., the angle between the foot and ground at initial contact) of the dominant leg (the preferred kicking leg) (Fu et al., 2017) by reviewing the obtained videos frame by frame (Heiderscheit et al., 2011). Consequently, all runners were rearfoot strikers. They ran a minimum of 15 km/week for at least 3 months prior to the study. Participants were excluded if they had any lower limb musculoskeletal injuries in the previous 6 months. No significant differences in age, height, weight and weekly mileage were observed between the two groups. This study was approved by the Institutional Review Board of the Shanghai University of Sports (No. 2017007). Informed written consent was obtained from each participant prior to their participation in this study.

Table 1 Demographics for participants.

Group	First visit
(n)	Second visit
(n)	Age
(years)	Height
(cm)	Weight
(kg)	Weekly mileage (km)	
Cadence retraining group	15	12	23.6 ± 7.5	174.8 ± 4.4	71.8 ± 4.9	23.3 ± 3.3	
Control group	15	12	23.7 ± 1.2	175.5 ± 5.1	70.8 ± 7.3	22.9 ± 4.3	

Instrumentation

A 12-camera motion capture system (100 Hz, T40; Vicon Motion Inc., Oxford, UK) was used to collect kinematic data. Ground reaction force data were captured by using two 90 cm × 60 cm × 10 cm force platforms (1,000 Hz, 9287B; Kistler Instruments AG Corp., Winterthur, Switzerland). The kinematics and ground reaction force data were simultaneously collected using the Vicon system. A Photogate system (Witty-Wireless Training Timer; Microgate Corp., Bolzano, Italy) was used to monitor over-ground running speed. Conventional running shoes (Nike Air Zoom Pegasus 34) were used by the participants during the experiments (Fig. 1A).

Figure 1 (A) Experimental shoes and (B and C) experimental set up.

Experimental protocol

The participants visited the laboratory twice, at baseline and at the end of the gait retraining program. Prior to data collection, the participants were required to wear uniform clothing, including a vest, socks and shoes, and walk for 2 min and run at 3.33 m/s on a treadmill for 5 min as a warm-up. Thereafter, a total of 40 markers were placed on the participants, and static calibration was performed. The anatomical locations of the markers were the right/left ilium crest tubercle, right/left posterior superior iliac spine, right/left femur greater trochanter, right/left anterior superior iliac spine, right/left femur lateral epicondyle, right/left femur medial epicondyle, right/left fibula apex of the lateral malleolus, right/left tibia apex of the medial malleolus, right/left head of the fifth metatarsals, right/left head of the first metatarsus and right/left posterior surface of the calcaneus (Fig. 1B). In addition, three tracking markers were placed on the thigh and shank. The participants were instructed to run over the ground across a 10 m runway (Fig. 1C) at 3.33 m/s during which kinematic and ground reaction force data were captured. The running speed was considered acceptable if the deviation was within 5%. Three successful running trials were collected for each participant.

Retraining protocol

All of the participants were required to run at their preferred speeds during the cadence retraining protocol (Hafer et al., 2015). Running speed and cadence during training were monitored using the commercial running application CODOON© (Chengdu Ledong Information Technology Co., Ltd., Chengdu, China). Each participant received a sport belt bag in which to place their mobile phones during running, and they were instructed to place the bag above their sacrum. The participants were asked to run outdoors three times (30 min/run) at a comfortable speed to determine their preferred speed and cadence. The preferred speed and preferred cadence were the average values obtained from three outdoor trials. Participants in the CAD group were instructed to run with a 7.5% increase in cadence, whereas those in the CON group ran without any change in cadence (Willy et al., 2016b). Participants in the retraining group were informed about and given access to a mobile-based metronome application with tempos set to a 7.5% increase in cadence. Figure 2 shows the cadence retraining protocol, which lasted for 12 weeks with three sessions a week and 5–48 min each session (Neal et al., 2018). Participants used their preferred running mode, namely, treadmill or over ground, to complete their retraining. The retraining protocol constituted part of the participants’ running volume so that their total weekly running volume remained unchanged. After each retraining session, participants could check their average cadence, speed and running volume on the CODOON© running application. They were also required to submit data recorded by the application to the researchers. Participants were excluded if their training protocols were interrupted more than three times or if their cadence did not achieve the targeted cadence for 3 weeks since the beginning of training. Weekly group trainings were provided three times a week in the CAD group to ensure compliance, and participants chose one of weekly group training sessions in which to participate on the basis of their schedule. During group training, the participants performed an 8 min warm-up, such as dynamic stretching, under the guidance of the researchers. Then, the participants began to run according to their retraining schedule. The participants did not receive guidance on running techniques because the weekly group training aimed to ensure compliance and the quality of retraining.

Figure 2 Cadence retraining protocol.

Data processing

The Visual 3D software (v5; C-Motion, Inc., Germantown, MD, USA) was used to compute the 3D kinematic and kinetic variables of the lower extremity during running. Marker trajectories were filtered with a cut-off frequency of 7 Hz via a fourth-order Butterworth low-pass filter (Yang et al., 2020). A seven-segment lower extremity model was built via the Visual 3D, and CoG was estimated from this model. Impact force variables included impact peak, vertical instantaneous load rates (VILR) and vertical average load rates (VALR). In rearfoot strike runners, impact peak was defined the first peak in the ground reaction force curve (Fig. 3B). Load rates was calculated on the basis of the method described by Futrell et al. (2018). In brief, a point of interest (POI) was defined as the first point above 75% of a participant’s body weight with an instantaneous load rate of less than 15 body weight/s. VALR (the average slope) and VILR (i.e., the maximum instantaneous slope) were then calculated from 20% to 80% and from 20% to 100% of the force at POI, respectively (Fig. 3B). We also calculated lower extremity stiffness (Liu et al., 2006), kleg, as shown in Eq. (1). Kinematic variables of the hip, knee and ankle joints included foot angle (the angle between the foot and ground) at initial contact (Fig. 3A) and peak joint extension and peak joint flexion angles during the stance. The times from initial contact to impact peak (tip), vertical velocities of the CoG at initial contact and vertical excursion of the CoG during the stance phase were also evaluated.

(1) kleg=GRFiΔy

where GRFi is the vertical ground reaction force at the lowest position of the CoG and Δy is the maximum vertical displacement of the CoG.

Figure 3 Scheme of (A) lower extremity kinematic and (B) impact force variables.

POI, point of interest.

Statistics

The mean and standard deviation for each variable was calculated. Two-way repeated measure ANOVA was used to characterise the effects of training (pre- and post-training) and group (CAD and CON) on each variable. Independent sample and paired t-tests were used as post-hoc tests when a significant interaction was detected to assess potential group effects between CAD and CON and retraining effects pre- and post-training, respectively. The observed effect size (η2) was considered in the ANOVA results, and effect size (Cohen’s d) was considered in the paired and independent sample t-tests results. The 95% confidence interval (CI) of the differences in group effects was reported. The criterion α level was set to 0.05. All statistical procedures were conducted using SPSS software (Version 20; SPSS, Inc., Chicago, IL, USA).

Results

Dropout rate

Thirty participants (15 in the CAD group and 15 in the CON group) completed the pre-training tests on their first visit to the laboratory (Table 1). However, in the CAD group, one participant was excluded because of insufficient training volume, and two participants withdrew for personal reasons or because their results showed more than three interruptions. In the CON group, two participants were lost to contact, and one participant withdrew for personal reasons. Overall, 24 participants, 12 in the CAD group and 12 in the CON group, completed the 12-week cadence retraining protocol and reported to the laboratory for post-training tests (Table 1). No significant difference in average running volumes was observed between the CAD and CON groups (CAD: 23.3 ± 3.3 km/week, CON: 22.9 ± 4.3 km/week).

Cadence and step length

Figure 4A shows a significant training × group interaction effect for cadence (P < 0.001, η2 = 0.867). Specifically, cadence significantly increased by 5.7% (161.3 ± 9.5 step/min vs. 170.5 ± 9.2 step/min) in the CAD group (P < 0.001, Cohen’s d = 3.87) but not in the CON group (164.3 ± 7.7 step/min vs. 165.5 ± 6.8 step/min, P > 0.05) after training. A significant main effect of training was observed for step length, which decreased by 4.1% (2.49 ± 0.16 m vs. 2.39 ± 0.14 m) in the CAD group after training (P = 0.011, η2 = 0.259) (Fig. 4B). No significant difference for step length was observed in the CON group (2.54 ± 0.16 m vs. 2.51 ± 0.14 m). Step length in the CAD group was 4.9% lower (2.39 ± 0.14 m vs. 2.51 ± 0.14 m) than that in the CON group after training (P = 0.04, 95% CI [−0.245 to −0.006], Cohen’s d = 0.94).

Figure 4 Effect of 12-week cadence retraining protocol on (A) cadence and (B) step length.

CAD, cadence retraining group; CON, control group; *indicates significant difference between pre-training and post-training in the CAD; #indicates significant difference between CAD and CON after retraining. P < 0.05.

Impact force variables

Figure 5A shows significant training × group interaction effects for impact peak (P = 0.022, η2 = 0.217). Impact peak significantly decreased in the CAD group (P = 0.003, Cohen’s d = 1.10) but not in the CON group (P > 0.05). Meanwhile, impact peak in the CAD group was significantly lower than that in the CON group after training (P = 0.038, 95% CI [−0.443 to −0.013], Cohen’s d = 0.95). Significant main effects of training were observed for VALR and VILR. Specifically, VALR (P = 0.029, η2 = 0.198) and VILR (P = 0.025, η2 = 0.209) decreased in the CAD group after training (Table 2).

Figure 5 Effect of 12-week cadence retraining protocol on kinematics variables.

(A) Vertical excursions of the CoG. (B) Vertical velocity of the CoG at initial contact. CAD, cadence retraining group; CON, control group; CoG, center of gravity; *indicates significant difference between pre-training and post-training in the CAD; #indicates significant difference between CAD and CON after retraining. P < 0.05.

Table 2 Effect of 12-week cadence retraining protocol on lower extremity biomechanics.

Variables	Cadence retraining group	Control group	
Pre-training	Post-training	Pre-training	Post-training	
Impact peak (BW)	1.86 ± 0.3	1.67 ± 0.27	1.88 ± 0.25	1.9 ± 0.23	
Time from initial contact to impact peak (ms)	31.29 ± 1.82	31.74 ± 2.29	30.13 ± 4.53	30.6 ± 4.56	
Vertical average load rates (BW/s)	91.59 ± 18.91	77.31 ± 15.12	92.47 ± 20.04	91.37 ± 25.02	
Vertical instantaneous load rates (BW/s)	108.78 ± 24.47	92.75 ± 18.49	109.88 ± 22.21	107.42 ± 25.82	
Lower extremity stiffness (kN/m)	34.34 ± 7.08	38.61 ± 6.51*	38.08 ± 7.35	38.36 ± 5.59	
Foot angle at initial contact (°)	18.27 ± 5.59	13.74 ± 2.82*	17.02 ± 6.54	16.97 ± 7.16	
Maximum dorsiflexion angle during stance (°)	20.10 ± 4.33	20.50 ± 3.91	20.63 ± 3.81	19.74 ± 4.6	
Maximum knee flexion angle during stance (°)	−38.60 ± 5.00	−36.50 ± 5.45*	−37.74 ± 2.78	−37.22 ± 4.42	
Maximum hip flexion angle during stance (°)	−14.98 ± 3.27	−14.70 ± 6.27	−14.39 ± 4.06	−13.10 ± 4.41	
Notes:

BW, body weight.

* Significant difference between pre-training and post-training in the CAD.

Kinematics and joint mechanics

Significant training × group interaction effects were observed for foot angle (P = 0.04, η2 = 0.178), vertical velocity of the CoG at initial contact (P = 0.035, η2 = 0.186) and vertical excursion of the CoG (P = 0.001, η2 = 0.409). Foot angle (P = 0.003, Cohen’s d = 1.09), vertical velocity of the CoG at initial contact (P = 0.002, Cohen’s d = 1.16) and vertical excursion of the CoG (P < 0.001, Cohen’s d = 1.83) decreased in the CAD group after training (Table 2; Fig. 5). Moreover, vertical excursion of the CoG in the CAD group was significantly lower than that in the CON group after training (P = 0.025, 95% CI [−0.015 to −0.001], Cohen’s d = 1.03). Significant main effects of training were observed for peak knee flexion angle. Specifically, peak knee flexion angle (P = 0.048, η2 = 0.166) was decreased in the CAD group after training (Table 2). A significant main effect of training was observed for lower extremity stiffness, which increased in the CAD group after training (P = 0.048, η2 = 0.166) (Table 2).

Discussion

This study aimed to characterize the effects of a 12-week cadence retraining protocol on impact peak, load rates and other lower-extremity biomechanical variables. Significant reductions in impact peak and load rates were observed in the CAD group. The preferred cadence in the CAD group significantly increased after 12-week cadence retraining, consistent with the results of previous studies conducted by Hafer et al. (2015) and Neal et al. (2018). However, the average change in preferred cadence in the present study was +5.7% between pre- and post-training. By contrast, the preferred cadence changes in the studies of Hafer et al. (2015) and Neal et al. (2018) were +2.4% and +7.6%, respectively, which were induced by increases of 10% and 7.5% in cadence during retraining. Compared to the study by Hafer et al. (2015), the cadence after training in the current study and the study by Neal et al. (2018) was closer to the prescribed cadence, which was likely due to the enhanced supervision in training. In addition, real-time feedback was provided in Neal et al.’s study, which may lead to the differences in cadence improvements between Neal et al.’s and present study.

In the present study, impact peak was significantly reduced by 10.2% in the CAD group after training, which was greater than the 7.6% decrease observed (pre-training vs. post-training) in the study of Hobara et al. (2012). Moreover, impact peak in the CAD group after training was 12% significantly lower than that in the CON group after training (CAD vs. CON). This decrease may be related to reductions in vertical velocity and vertical excursion of the CoG (Derrick, Hamill & Caldwell, 1998). According to the impulse–momentum principle, impulse is equal to the change in the body’s momentum. During running, the momentum exchange between the ground and a portion of the body when it comes to a full stop causes an impact peak (Addison & Lieberman, 2015). In the present study, the vertical velocity of the CoG at initial contact was significantly decreased in the CAD group after retraining, but no difference in tip was observed between pre- and post-training. This finding may indicate that the observed decrease in impact peak in the CAD group after retraining may be due to the decreased vertical velocity of the CoG at initial contact after retraining. The foot angle, which reflects the foot strike pattern during running, significantly decreased with increasing cadence (Allen et al., 2016; Heiderscheit et al., 2011). In addition, a lower impact peak has been found in a smaller foot angle at foot contact with a rearfoot strike (Mercer & Horsch, 2015). In the present study, the foot angle in the CAD group significantly decreased by 4.5° after retraining, which may partially explain the decrease in impact peak in the CAD group after retraining.

VALR and VILR in the CAD group were significantly reduced after retraining, consistent with the findings reported by Hafer et al. (2015) and Willy et al. (2016a). Lieberman et al. (2010) found that the load rates was lower in forefoot strikes than that in rearfoot strikes. In the present study, decreased foot angles in the CAD group after retraining slightly altered the strike pattern, which may contribute to reductions in load rates. Injured runners were reported to have higher load rates than non-injured runners in a prospective investigation (Davis, Bowser & Mullineaux, 2016). Therefore, the decrease in load rates after retraining indicate that cadence retraining may reduce the risk of running injuries, such as stress fractures.

The knee joint was highly sensitive to changes in cadence during the stance phase (Heiderscheit et al., 2011). 5.7% (9.2 step/min) increase in cadence induced significant changes in peak knee flexion angle (Dos Santos et al., 2016; Neal et al., 2018). The increase in cadence in the CAD group decreased the peak knee flexion angle and vertical excursion of the CoG; no significant differences were observed for the hip and ankle joint angles between pre- and post-training. Additionally, lower extremity stiffness significantly increased in the CAD group after training, which may be due to the reduced vertical excursion of the CoG during the stance phase induced by the decrease in peak knee flexion angle.

Some limitations of this study must be considered when interpreting the results. Firstly, all of the participants were male; whether females would show the same effects after 12-week cadence retraining remains unclear. Secondly, the running biomechanics obtained from a limited run-up (10 m) with a relatively small area (60 cm × 90 cm) for foot placement may slightly differ from that obtained during outdoor over-ground running. Moreover, long-term retention effects caused by retraining changes were not evaluated in this study. Finally, whether the training effect will maintain when individuals reach fatigue is unknown, and should be considered in future studies.

Conclusion

Twelve-week cadence retraining significantly increased runners’ cadence by 5.7%. The increased cadence effectively decreased a number of impact force variables, namely, impact peak, VALR and VILR. Given the close relationship between impact force variables and running injuries, increasing the cadence as a retraining method may reduce the risk of some impact-related injuries. A decrease in foot angle at initial contact after training may provide a mechanical explanation for the observed decrease in impact force variables. Furthermore, the vertical excursion of the CoG decreased, thereby increasing lower extremity stiffness. Hence, cadence retraining can lead to lower extremity biomechanical changes.

Supplemental Information

Supplemental Information 1 Raw data was exported from the mean and standard deviation of the cadence retraining group and control group between pre-training and post-training.

Click here for additional data file.

Additional Information and Declarations

Competing Interests

Author Contributions

Human Ethics

Data Availability

The authors declare that they have no competing interests.

Junqing Wang conceived and designed the experiments, performed the experiments, analyzed the data, prepared figures and/or tables, authored or reviewed drafts of the paper, and approved the final draft.

Zhen Luo performed the experiments, authored or reviewed drafts of the paper, and approved the final draft.

Boyi Dai analyzed the data, authored or reviewed drafts of the paper, and approved the final draft.

Weijie Fu conceived and designed the experiments, authored or reviewed drafts of the paper, and approved the final draft.

The following information was supplied relating to ethical approvals (i.e., approving body and any reference numbers):

The Institutional Review Board of the Shanghai University of Sports approval to carry out the study within its facilities (2017007).

The following information was supplied regarding data availability:

The raw measurements are available in the Supplemental Files.

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
