# Peer review of "Effects of 12-week cadence retraining on impact peak, load rates and lower extremity biomechanics in running"

_PeerJ, doi:10.7717/peerj.9813_

## Round 0.1 · original submission · Major Revisions

Three reviewers have now considered your paper and have identified a number of major concerns. Regardless, I would like to give you the option to revise the paper if you think you can address the concerns raised.

I look forward to receiving a revised manuscript should you decide to undertake the additional work required.

Reviewer 1 ·

Basic reporting

See general comments

Experimental design

See general comments

Validity of the findings

See general comments

Additional comments

The study assessed the effect of cadence retraining on impact related variables and lower limb kinematics and kinetics. Overall the study is well written and examines an important research question. I have suggested some changes to improve the reporting of the manuscript. My main concern relates tot the presentation of results, where I am unclear as to what statistical tests have been used and reported. Please see specific comments in the points below.

1. Abstract – it would be nice see some data reported in the results section of the abstract. If you are tight on word limit perhaps present the most important results. Please see my comments in relation to statistical analysis and results as to what is appropriate to include in the results.

Introduction
2. Line 44 – please change patellofemoral joint pain synthesis to patellofemoral pain – as that is the current description of this condition. Also injuries to the lower leg have bene reported to be just as common as the knee in more recent publications. Please document this (e.g. Buist et al. BJSM 2010: 44:598-604; Franke et al. JOSPT 2019: 518-529).

3. Lines 46-52: In this section please differentiate those studies that have shown an association with injury (i.e. retrospective studies) to prospective studies which provide a greater level of evidence. In other words, is there prospective evidence for excessive impact forces and loading rates as a risk factor for running injuries?

Methods
4. Line 99 – you have indicated sampling rate twice in this sentence – please remove one duplicate.

5. Lines 111-118: Please provide more details in regard to the sequence of your experimental protocol. For example, were markers placed on participants and static calibration performed before or after the 5 min run on the treadmill?

6. Line 132: The Alfredson reference seems to be the incorrect reference here

7. Lines 135-137: How did the researchers ensure compliance to the retraining protocol? You mention that cadence was uploaded, but what was the action if participants were not achieving the desired cadence?

8. Line 138: please provide details on the weekly group training.What did you do in these sessions?

Results
9. Overall for your results you should be presenting the interaction effect as your main finding, not the time effects. If there was truly an effect of gait retraining on your variables there should be group differences at follow-up. Reporting changes over time as treatment effects are misleading. I am little confused as to how you present these group differences, for example is the p-value reported for impact peak the interaction effect, or the result of the independent t-test? You have reported in your statistics that independent or paired t-tests were used when a significant interaction was found, yet you haven’t reported where interactions were found, so what is the justification for following up with these tests? As per your methods if you did not find interaction effects you should not be reporting these post-hoc tests.

10. It would be useful if you could also present the 95% confidence interval of the differences for group effects (i.e. impact peak, step length, CoG excursion). Also please provide some magnitude of effect for the difference (i.e. ES)

11. Did you not find an interaction effect for cadence? The figure indicates that it may have been close to significance? Given this is your intervention it is surprising there was no group differences at follow-up.

Discussion
12. As per my comment in the results, overall in the discussion you need to indicate that only peak impact forces, step length and C0G excursion exhibited a group by time interaction. Thus these are the only variables that demonstrated differences related to time and group allocation. Much of your discussion is focused on variables that showed a time effect and this should be reported with some caution.

13. Line 242: you still have not indicated how you controlled compliance to the cadence retraining so find it hard how you can suggest this is a reason for the reported differences. Please provide details in the methods I regard to how you ensured compliance with the increased cadence.

14. Line 245: In the results you indicate a reduction of 12% in peak impact forces, which is different to that reported here.

Line 307: You need to temper your conclusions based upon my comments related to your statistical 15. analysis and presentation of results.

Reviewer 2 ·

Basic reporting

The authors are highlighting an important topic in running injury research especially gait retraining with emphasis on cadence retraining. There is evidence in the literature demonstrating that ( Willy et al.2016, Hobara et al. 2012 ) increasing cadence above one’s natural step rate appears to result in an immediate reduction in load rates. Willy et al. 2016 used a wireless accelerometer as a cue for 8 training sessions, to increase preferred step rate by 7.5% and this was effective at reducing impact forces, peak hip adduction and eccentric knee joint work in healthy runners.Hafer et al. also showed that that increased running cadence using cadence retraining reduces kinematics and kinetics that have been tied to overuse running injuries.
The introduction lacks clear justification for performing the study. The authors do not provide any deficiencies in the running literature to justify a need to do this study.
There is no clear distinction between increase in cadence or decrease in step length that is the focus of the study. Hypothesis is not specific and focused enough.
There is no justification for the amount of variables that were chosen for analysis in this study (kinematics and kinetics)

The raw data and the figures were helpful to understand the data.
The description of the data in the results in the data was muddled because of the lack of clarity if it was talking about the data within the CAD group or CON group (pre and post) or comparing the CAD and CON group.

Experimental design

There is no discussion about sample size calculations.
There is no information about how the randomization was performed for the two groups. If any of the investigators were blinded and who performed the randomization.
The marker set used for the gait analysis was not described.
There is no justification for using the 12 week retraining period or the 7.5% increase in cadence. They have cited a study but why they felt this protocol was appropriate for their study was not stated.
It is not clear why the runners ran on the treadmill prior to data collection .
Were three running trials enough to collect enough data points ? It wasn't mentioned for how many seconds every trial was performed. The assessment was performed at fixed speed but retraining occured at preferred speeds.The authors do not provide details of the weekly group training sessions .
There were many(atleast 17) biomechanical variables analyzed in this study and except the impact forces or loading rates there was no justification for the rest of the variables. Which plane were the peak moments of hip, knee and ankle analyzed or was it an overall support moment?
Some variables like vertical momentum at initial contact,maximum extension/dorsiflexion angle during the stance phase;maximum flexion/plantar flexion angle during the stance phase;centre of gravity,sagittal plane distance between ankle and knee/ ankle and hip at initial contact are variables which have not been previously reported in the running literature. The validity of the overall findings is questionable without knowing the impact of these variables on injury status in runners. I am not sure if centre of gravity is what the authors use for centre of mass which is used in the running literature.

T-tests to look at differences between the two groups were not performed. Height of the participant has been shown to impact step length.

Validity of the findings

The authors do a good job using tables and figures to describe the data. The discussion section introduces the role of foot strike pattern which tends to shift the focus from cadence retraining . Previous literature has highlighted the role of forefoot strike pattern and decrease in average and peak loading rates( healthy runners and runners with tibial stress fracture)(Boyer et al., 2014; Shih et al., 2013,Yong et al.2018; Futrell et al.2018)
The novelty of this study is not clearly demonstrated by the authors. It rather seems like a validation of other studies or a replication of the protocol.

Additional comments

Abstract:
Line 17: Please use Injuries not injury status
Line 19: during running
Results section: This section needs numbers and p-values. It is very wordy.
Conclusion: Needs to be more impactful and not be similar to the introduction.
Introduction:
Line 36: Please provide the name of the report
Line 40: Please ensure that the citation is accurate. Typically, all running studies reference the RunningUSA.org for the statistics on participation
Line 44: Patellofemoral pain syndrome
Line 47: Consider using the word Impact rather than affect
References were nor provided for: Line 40, Line 49
Line 48-49: Please make sure you change it to running injuries everywhere rather than running injury status
Please rewrite the sentences for grammar corrections:
Lines 52: compared “ to “ healthy runners
Lines 58: decreased
Lines 59: indicate
Lines 59-65: There is no clear distinction between increase in cadence or decrease in step length that is the focus of the study.
Lines 73-77: There is no justification for the amount of variables that were chosen for analysis in this study (kinematics and kinetics)
Lines 81-83: The hypothesis needs to be clear and specific
Lines 85-95: Please perform sample size calculation and describe the randomization protocol. Was there any blinding of the investigators who performed data analysis and collected the data? Please specify if it was lower limb musculoskeletal injuries.
Line 100: Please describe which marker set was used.
Line 104: Please remove the word force before ground reaction force data
Line 112: Please describe why you had them run on treadmill
Retraining Protocol: Please describe why you did not collect data at preferred speed if they ran for 12 weeks at their preferred speed. What was the difference in speeds ? Decrease in running speed can lead to decrease in kinetic forces and viceversa. Please describe the group training sessions
Data processing: Please justify your reasons for selecting so many variables for analysis. There are many variables which are not commonly used in the running literature(vertical momentum at initial contact,maximum extension/dorsiflexion angle during the stance phase;maximum flexion/plantar flexion angle during the stance phase;center of gravity,sagittal plane distance between ankle and knee/ ankle and hip at initial contact ). Does Center of gravity refer to center of mass? Which plane were the peak moments of hip, knee and ankle analyzed or was it an overall support moment?
Please perform t-tests for demographic variables between two groups. Height of the participant can impact step length.
Line 192-193: Revise for grammar.
Line 220,223,225: which group?
Line 240: Please explain why you think your study did not see the increase in cadence that you expected( 5.7% vs 7.5%) specifically if any biomechanical adaptations can explain this
Please link your results to cadence retraining and kinetic parameters linked to running injury rather than focusing on foot strike pattern. Can you describe how the results of the study can be used for future studies especially in participants with running injuries.

·

Basic reporting

I commend the authors on undertaking and submitting an article on a very clinically relevant topic which should be of interest to readers.

The article examines the effect of a 12-week cadence retraining program on impact forces and lower extremity biomechanics. Whilst the authors' study findings do support the original research goals, their conclusion that reducing impact forces may decrease the risk of impact-related running injuries needs further exploration in the introduction. The link between running injuries and impact forces remains unclear which needs to be indicated.

The article is well written although some sections, including tables and figures, need further detail or clarification and I have included specific comments in the General Comments section below which I hope the authors will find helpful.

Experimental design

The research is within the Aims and Scope of the journal.

However, further elaboration of the introduction to the topic is required to justify the study and research question. Additionally, the link between impact forces and running related injuries needs to be explored further to justify any of the conclusions made in the abstract. The study primarily attempts to answer whether increasing running cadence can decrease loading rates in runners and should likely focus on this. There are additional studies, such as Yong et al. (2008) and Chan et. al (2018) which have reviewed the impact of gait retraining, including increasing cadence, on loading rates in runners and should be referenced within the article. This will strengthen the authors' findings and can help to justify the association with injury reduction.

Additionally, the methodology requires additional detail. There should be further justification regarding the chosen sample size and elaboration of the participants running experience and training history, where possible. This will assist with the application of the research findings.

Validity of the findings

There are some concerns regarding the methodological quality in terms of the reliability of assessing running gait over a force platform where participants have to place the foot. Further information on the experimental set-up would help.

Additionally, the validity and reliability of CONDOON app and use phones to measure running metrics will need to be included in the study to justify the findings.

Additional comments

INTRODUCTION
Line #42-43: “In addition, 19.4% to 79.3% of long-distance runners experienced 43 injuries” – Can you further elaborate why the large variance; for example, novice runners or those who run increased distance show higher injury rates.
Line #42-43: “whilst 63% of them had a history of lower extremity injury” – This is unclear who it is referring to. There is no reference to this in van Gant et. al. 2007
Line#44: “such as patellofemoral joint pain synthesis” – Remove word “synthesis”
Line#49: Can please you reference running retraining studies
Line#55: “Increasing the running cadence or decreasing the step length” - Remove “the”
Line#56: “at a certain speed” – What speeds
Line#58-59: “decreased” – Grammatical error. Changed to past tense
Line#60: “the cadence or reducing the step length” - Remove “the”. Also, increasing cadence and reducing step length are the same.
Line#66: “In regard” – Change to in regards
Line#67: Is 6 weeks defined as a “long term change”
Line#68-72: “Hafer et al. (2015) did not find significant changes in impact forces in another cadence retraining intervention, but the loading rates decreased after 6-week cadence retraining with a 10% increase in cadence. In addition, reductions in peak loading rates and average loading rates were observed after an eight-session cadence retraining intervention with a 7.5% increase in cadence”
- Consider re-wording this clearer and separate the two studies.
Line#72: “cadence retraining modifies” - If it decreased say it decreased loading rates
Line#75-77: “Assessing these additional variables may result in better understanding of cadence retraining on running biomechanics and its associated risk of injuries”
- Are you assessing the risk of injury? You need to elaborate on the link between impact forces and running injuries.


METHODS
Good description
Line#89: How did you determine that participants were rearfoot strikers?
Line#102-103: Rewrite as 90cm x 60cm x 10cm
Line#111: Remove “that is”. Write at baseline or commencement and at the end of the program.
Line#115: I would be interested to see the experimental design in the appendix with a description of the run-up and force plate location. This is a limitation of the study as you are assessing running biomechanics with a limited run-up with a relatively small area for foot placement.
Line#121: How did you determine “preferred speed”
Line#123: Need to provide some data regarding the reliability and validity of CODOON running app in tracking running volumes, speed and cadence. How accurate are phones in measuring cadence? Have you controlled for the positioning of the phone and where will it be held during trials? ie. Held in hand or strapped to the arm.
Line#124” “The participants were asked to run outdoors for three times” – remove word “for”
Line#126 – Grammatical. Consider rewriting
Good summary of data processing and statistics. Recommend further statistical assessment prior to publication to validate the analysis.

RESULTS
Line#191-192: Consider summarising more clearly.
Line#197: “CAD: 23.3+/-22.9 km/week” – There appears to be an error in standard deviation calculations. I could not validate these numbers as only patient training characteristics at baseline were reported in the supplementary material. If not, can you explain the large variance in the training of the CAD group?

DISCUSSION
Consider elaborating some of the arguments in the introduction rather than introducing new arguments in the conclusion.

Lines#247-251: These arguments should be included in the introduction and can be referred back to in the discussion.
Lines#258-259: Consider re-wording
Lines#269-271: Do we have evidence that reducing loading rates can decrease injuries? Yong et al. (2008) “Acute changes in foot strike pattern and cadence affect running parameters associated with tibial stress fractures” has also explored this and should be referenced in the text.
Lines#275-277: Needs reference. Additional grammatical errors. Consider revising sentence.
Lines#296-297: “Although cadence increased by 5.7% in the CAD group after 12-week retraining, the 297 magnitude was only 170.5 step/min, leaving additional space for increasing cadence”
- You have not reported on normative cadence values for runners. Why is there additional space? Also is there a ceiling effect or where increasing cadence can have a negative effect or reduce running economy. One participant increased to a cadence of 193 steps/minute.

CONCLUSION
Lines#309-311: “Increasing the cadence as a retraining method may reduce the risk of impact-related injuries” - Again, needs further link in the introduction to justify this.
Lines#310-312: “Meanwhile, the foot became “flatter” when its position was closer to the CoG at the initial contact after training, providing the mechanical explanation for the decreased impact forces” Elaborate on the mechanical explanation. Also, consider the change in vertical oscillation (excursion).
Lines#312-315: “Hence, cadence retraining can lead to the biomechanical adaptation of the entire lower extremity to the running pattern” – I am unsure what you are trying to get across here

REFERENCES
The references have some consistent errors. Consider removing DOI unless required.
Lines#331 and 408 have minor author errors.

TABLES AND FIGURES:
Table 1:
Elaborate or use key/legends groups (CAD and COG)
Label standard definitions
Table 2:
Consider using a line chart to illustrate
Table 3:
Variables are not labelled on the chart and it is difficult to refer to the note below. Consider full label on the table and in landscape mode.
Needs consistent use of decimal places
Need to label standard deviations
Figure 2-5:
Need to elaborate abbreviations (eg. POI Lah)
Consider expanding CAD and CON

---

## Round 0.2 · Major Revisions

Thank you for your revision of this paper. As you can see there are still a number of major considerations that have been raised by the reviewers.

In addition to the scientific considerations raised by the reviewers, please also ensure you have your paper thoroughly proofread to ensure it is free of typographical errors. This is your responsibility and not that of the journal, failure to do so may result in a rejection of the paper.

Thank you.

Reviewer 2 ·

Basic reporting

My overall comments about the Introduction: You haven’t highlighted the need for the study, You just summarized the available literature and It seems like because lower extremity biomechanics has plethora of variables available for assessment, you are picking the one’s that have not been highlighted. I mentioned it in first review , please justify the need for the study and I feel this still lacks that.

Experimental design

The calculation of variables and statistical tests performed are valid.
My suggestion would be that If you still want to include all your variables( kinematics and kinetics) could you use linear regression techniques to evaluate which variables predict impact forces and loading rates?

Validity of the findings

The Discussion revolves around Impact forces, Loading rates , foot angle and some kinematic variables. In my opinion, It should focus more on mechanisms of changes and differences between the two groups . There is no mention about changes in control group and the differences seen with CAD group.

I would recommend rewriting the limitations.

The conclusion was strong and focused on the big picture points from the study and that is why I would recommend rewriting the paper to focus on the changes in impact forces, loading rates rather than including all the 30 variables.

Additional comments

Thank you for your resubmission and for addressing the comments. It is more clear now with what the study intends to do but I am still not sure about the need to include 30 variables ( kinematics and kinetics) to evaluate the changes of a 12 week cadence retraining program. In your Introduction, you suggest " insufficient evaluation of variables and small sample size of previous studies as the need to perform this study but your study also has a small sample size and you did not address why all the variables you included in your study are important to evaluate.
I have made specific comments below but I would recommend the focus of the paper be strictly on the kinetic variables(Impact forces and loading rates and how CoG and foot angle play a role to support your findings) You should also highlight your 12 week retraining program and how the compliance tracking and meeting participants was the novel part in this study.

Specific Comments:
Abstract:
Lines 16-17 : You only focus on Impact forces and vertical loading rates and then lines 18-19 you talk about Impact forces and lower extremity biomechanics. I would recommend being consistent both the times
Lines 22: Please describe if the control group ran without any changes in cadence
Lines 24: Please mention in the Cadence training group so we know you are talking about this group. If in the results you are including the data for some variables then you have to include it for all variables especially if all of them are significant.
Line 31: Grammar, it should be knee range of motion.
Line 33: I would include the % increase in cadence in the sentence.
Line 34: Grammar, I would recommend using induce not induced. I would also recommend using the word “leads to” instead

Introduction:
Lines 41-43: 5.83 million runners vs 50 million recreational runners Is a huge difference in participation numbers. Can you cite the either the numbers for participation in running events or recreational runners every year for both the countries.
Lines 43-45: I would consider removing the sentence on running has more injuries than other activites, it sounds redundant. You have already provided injury numbers.
Lines 43-46: I would move this sentence to after the Messier reference. In addition, 19.4% to 79.3% of long-distance runners experienced lower extremity injuries (van Gent et al. 2007).
Line 48: I would recommend not using the word in recent publications. Instead use literature or there is data to suggest
Line 51: Grammar, I would recommend using influence not impact
Line 52-53: I would recommend using gait retraining studies instead of in running retraining strategies
Lines 57-60: Please rewrite this sentence to indicate specifically what is the point that you are trying to make
Lines 60-66: You talk about Impact forces, then you jump in to energy absorption then you talk about Impact forces again till line 70. Line 70-75 you talk about in general all effects of increasing cadence . I would recommend bringing everything together and justifying why al these things are important and how are they related to your study. in my opinion,listing all changes in gait variables doesn’t support your paper, it seems like you are doing a review of literature rather than telling the reader why these variables are important.
Line 69: Be consistent using terminology: stick to decrease step length or reduced step length. They mean the same thing but it’s important to be consistent.
Line 76: I would take out the word long term
There is confusion if you are focused on impact forces or the impact phase of running or both
Lines 83-86: I would specificially list how many participants were there in those studies and why you think that is a small number especially considering your study had 12 in the experimental group. Why do you think 6 weeks is a short training intervention time? Do you think it is practically feasible to retrain someone’s gait for more than 6 weeks and is there literature to support that you need more than 6 weeks for gait retraining to occur? Please explain what do you mean by Insufficient evaluation of variables.
Please rewrite these three lines. It is a very long sentence and it’s not clear what is that point that you are trying to make.
Lines 86-88: Please explain why you need more variables to understand cadence retraining and injury risk. Do you think the variables that you have listed from lines 60-75 are not enough ? Are there any variables linked to Injury that these papers don’t talk about ?
Lines 89: You haven’t talked about why 12 weeks is important.

Methods:
Lines 98-99: Why did you choose effect size 0.6?
Lines 102: Which Simple randomization procedure
Lines 103-106: Which limb did you calculate the foot angle ? Can you include the mean of the foot angle ?
Line 127 : What is test vest?
Line 138: Add “ protocol “ after cadence retraining
Lines 145-146: how did the runners know they were running with a 7.5% increase in cadence?
Lines 147-149: Rewrite this sentence
Lines 157-163: It is not clear what is the purpose of the weekly retraining group? Do you think it is feasible to expect runners to come in every week and why did you choose dynamic stretching and what muscles were targeted?
Lines 169: Why did you choose 7Hz for filtering the markers?
Results
Lines 226-Lines 233: I would recommend using the values and changes in cadences and step length in this paragraph
Line 240: Please change to CAD group “were” significantly

Discussion:
Lines 278: Change from in the present study was to “ were”
Neal et al. observed higher cadence post training ( 7.5% compared to your study 5.7%) Please talk about why you think there was a difference
Line 305-308: I would consider removing this because your study did not look at Injured runners and this doesn’t support your study. Instead focus on cadence retraining and impact on loading rates and mechanistically explaining how cadence retraining changes these variables.
Lines 320-321: How much was the change in cadence, please be specific.
Lines 320-334: I would recommend talking how your study contributed to the literature rather than restating the literature. I would focus more on mechanisms because its discussion section rather than highlighting the literature.
Lines 336: This is unclear. Do you mean step frequency ? That is not magnitude. This is the first time you mention step frequency. Also, there are no norms in the literature to suggest numbers for preferred step frequency.
Lines 338: Please explain step by step approach especially after you suggest the 12 week training improved compliance and had a longer retraining phase compared to other studies.

·

Basic reporting

Numerous grammatical errors remain within the manuscript.
Some examples are: Line #46: "bene", Line #63: "to related", Line #64 “CoG” Abbreviation but on 1st reference. I would recommend a full 3rd party grammatical review prior to publishing.

There are some areas that still require reference updates.
Line #41: "2017" - Needs updated reference.
Line #55-56: These are case series so please indicate accordingly.

The figures and tables are significantly improved.

Experimental design

I thank the authors for expanding on their methodological design.

A few smaller concerns are as follows;

Line #64. Derrick used CoM as a description, which is common with biomechanical studies on running. It would be helpful to see how you determined or estimated CoG. What this from a hip marker? If so, please include.

Line #124: The anatomical locations of the marker set could be described further in the appendix ao that the protocol could be reproduced.

Line #140: It remains unclear why 7.5% was chosen. This needs further justification for why it was chosen over 10%. Referencing other studies provided support but you still need clear reasoning.

Line #172. The impact variables are well described. However in Line #229 "impact forces" is used but it is unclear which variable is being referred to. This theme continues in the results section (Lines #231-234) and discussion (Lines #278)

Validity of the findings

There is one conclusion that needs further refinement.

Lines #281-283. "This finding indicated that the change in the vertical momentum of CoG was reduced during impact attenuation" - This conclusion appears incorrect. Velocity would not change as it was measured at initial contact. Any decreases would more likely be due to less centre of mass displacement.

Additional comments

The authors have done a good job answering the reviewers' concerns and amending the manuscript.

This shows a significant improvement in many areas. Readability may be improved with an external grammatical review as indicated above.

---

## Round 0.3 · Major Revisions

Thank you for revising the previous version of your paper. There is one reviewer in particular that still has some reasonable concerns with your paper. Please pay careful attention to these and provide a considered response to address all their concerns. Once this is complete a decision about the suitability for publication can be made. Thank you.

Reviewer 2 ·

Basic reporting

The authors have done a good job answering the reviewers’ concerns and making changes to the manuscript. There is an improvement in many areas especially with the Introduction to focus on kinetics( impact forces and loading rates) and highlighting why they chose the 7.5% increase in cadence and the 12 week cadence retraining period.
I am still concerned about including other lower extremity variables(kinematics) in the paper.Please change average and maximum loading rates to vertical instantaneous load rates (VILR) and vertical average load rates (VALR) everywhere in the manuscript.
I am also concerned about the discussion section as some of the points have still haven't been changed or explained from previous submissions.

Experimental design

Methods section was adequate . Please check my general comments for minor changes.

Validity of the findings

Please review my general comments for the discussion and conclusion section.
I am still concerned that your discussion and conclusion doesn't highlight what your study found and how it links to the results.
Please focus on your study rather than only citing previous research and those results.

Additional comments

Abstract:
Please include results from the control group also in the results.
Please rewrite the conclusion to reword lines 37-39

Introduction
Line 42 feels incomplete. I recommend adding form of physical activity or exercise in China and across the world.
Lines 43-44: Change the grammar. 5.83 million participants ran in 1,581 marathon events in China in 2018 . I would just check to make sure 1581 marathon events or races that include all distances( 5k,10k,half and full marathon)
Line 69: I would remove the word "immediate"
Line 71: Change "In regard" to "With regards to"
Line 82-83: Please elaborate on what do you mean by " reduce risks of rapid transition"
Line 83-85: I would recommend deleting lines 325-326. It sounds repetitive
Lines 87-88 are also a repetition from lines 75-78. I would recommend moving those lines here.
Methods
Line 103: Please change "was included " to "were included"
Line 106: Please change "running" patterns to "foot strike" patterns
Line 109: Please take out the word "visibly"
Line 129: Please add "gait retraining" program
Line 137: Please change " metatarsus" to "metatarsals"
Line 180-185: Futrell et al.2018 calculated vertical instantaneous load rates (VILR) and vertical average load rates (VALR). Please ensure that you mention these specific terms.
Line 185: Please add " We also calculated " instead of kinetic variables because impact peak and loading rates are also kinetics
Lines 187-188: Please only mention flexion and extension. All lower extremity joints are not dorsiflexion and plantar flexion, only ankle joint is.
Line 197: Please change "were" to "was" or change the sentence to " The mean and standard deviations were calculated for all variables"
Line 222-223: Please report changes if any in cadence in control group
Line 225-226 : Please indicate the values of step length in control group
Lines 230-237: Please refer to my earlier comments about terminologies
Lines 256-257: Please align this with study aims
Lines 258: Please remove "cadence retraining could decrease running impact force variables"or explain how you came to this conclusion.
Lines 264-266: Please reword this to explain why Neal et al.2018 had greater improvements in cadence compared to Hafer at al. 2015 and present study.
Lines 267-270: Please reword this to clearly explain your study variables VS Hobara et al. 2012 study values. It's confusing when you say impact peak decreased by 10.2% but then you mention it decreased by 12% in the CAD group compared to the control group.
Line 278: Please remove "recently"
Line 278-282: This section is muddled. What is subtle heel strike. What do you mean by change from rear foot to non-rearfoot strike. How is that subtle. Please provide values of the change in rear foot angles
Lines 285-293: Please refrain from using terms like " Positive effect on reducing risk of running injuries" and highlighting other studies that don't focus on gait retraining and its effects on loading rates and impact peaks.
Lines 294: Please change "touchdown" to "initial contact"
Line 295-301: This is repetition because you already included changes in foot angle in the earlier section. The study is not focused on foot strike pattern and or foot angle. Please reword this.
Line 302: please provide a reference
Lines 302-303: Please mention specific change in cadence
Lines 304-306: Please reword this to " The increase in cadence in the CAD group decreased the peak knee flexion angle and vertical excursion of the CoG.
Lines 307-308: Please remove this sentence "Thus, reduction in the vertical excursion of the CoG may be mainly attributed to the knee joint" or provide a justification based on your study results why you think this is the case, just because peak knee flexion angle was different doesn't justify that it is the reason why change in vertical COG was observed.
Lines 309-310: Your study did not " acutely increase the cadence" Your study was a gait retraining study over 12 weeks.
Lines 327-329: Please remove this sentence because your discussion did not look at vertical velocity of Cog and impact forces. Your discussion focused only on change in foot angle. Also, please refrain from including new terms, please be consistent with terminologies. Through out the manuscript you mention rear foot, non-rearfoot , subtle but now you include " flattening"

Table 2: Please include pre and post values of impact forces and load rates in Table 2
Lines 371: Futtrell reference has been deleted from the manuscript reference section. Please add it again.
Lines 390: Han T. 2019. Marathon statistics in China. THIS REFERENCE IS INCORRECT. PLEASE CITE IT ACCURATELY

·

Basic reporting

See General comments

Experimental design

See General comments

Validity of the findings

See General comments

Additional comments

I commend the authors on accepting the feedback and for making significant improvements to their article. As mentioned previously the article is on a very clinically relevant topic which should be of interest to readers. Well done.

I have a few minor suggestions to improve the paper prior to accepting.

Line #45. 2019 is not a reference. Update this to the author (Running USA).
Line #129. Remove i.e.
Line#289. "runners with high loading rates are more likely to develop injuries compared with those with low loading rates" We don't know this. One study is retrospective (Davis). The other reports symmetry angle of impact peak to be different between injured and non-injured runners. Consider revising this line but I think the argument is logical.
Line#298-9. "+10% of their preferred cadence". "+30% of their preferred cadence" Is this and increase?

---

## Round 0.4 · accepted · Accept

Thank you for your considered responses to the reviewer comments. I am now comfortable that all concerns have been appropriately addressed and that the paper is now of publishable quality.